# RNAi Technology: A New Path for the Research and Management of Obligate Biotrophic Phytopathogenic Fungi

**DOI:** 10.3390/ijms24109082

**Published:** 2023-05-22

**Authors:** Isabel Padilla-Roji, Laura Ruiz-Jiménez, Nisrine Bakhat, Alejandra Vielba-Fernández, Alejandro Pérez-García, Dolores Fernández-Ortuño

**Affiliations:** 1Departamento de Microbiología, Facultad de Ciencias, Universidad de Málaga, 29071 Málaga, Spain; ipadilla@uma.es (I.P.-R.); laura110493@uma.es (L.R.-J.); nisrinebakhat@uma.es (N.B.); alejandravielbafdz@gmail.com (A.V.-F.);; 2Instituto de Hortofruticultura Subtropical y Mediterránea “La Mayora”, Universidad de Málaga, Consejo Superior de Investigaciones Científicas (IHSM-UMA-CSIC), 29071 Málaga, Spain

**Keywords:** RNA interference, VIGS, HIGS, ATM-HIGS, dsRNA, SIGS, transgenic plants, powdery mildew, rust

## Abstract

Powdery mildew and rust fungi are major agricultural problems affecting many economically important crops and causing significant yield losses. These fungi are obligate biotrophic parasites that are completely dependent on their hosts for growth and reproduction. Biotrophy in these fungi is determined by the presence of haustoria, specialized fungal cells that are responsible for nutrient uptake and molecular dialogue with the host, a fact that undoubtedly complicates their study under laboratory conditions, especially in terms of genetic manipulation. RNA interference (RNAi) is the biological process of suppressing the expression of a target gene through double-stranded RNA that induces mRNA degradation. RNAi technology has revolutionized the study of these obligate biotrophic fungi by enabling the analysis of gene function in these fungal. More importantly, RNAi technology has opened new perspectives for the management of powdery mildew and rust diseases, first through the stable expression of RNAi constructs in transgenic plants and, more recently, through the non-transgenic approach called spray-induced gene silencing (SIGS). In this review, the impact of RNAi technology on the research and management of powdery mildew and rust fungi will be addressed.

## 1. Introduction

RNAi is a biological mechanism in which short noncoding RNAs (sRNAs) are used to deliberately downregulate gene expression at the transcriptional or posttranscriptional level. Posttranscriptional gene silencing is a tightly controlled system that relies on a group of proteins to coordinate gene silencing based on sequence complementarity between sRNA and target mRNA [1,2]. MicroRNAs (miRNAs) and short-interfering RNAs (siRNAs) are two types of regulatory sRNAs encoded by plants. miRNAs are 20–22 nucleotide (nt) sequences formed from a single-stranded RNA molecule that folds back on itself, creating a double-stranded region with a loop called RNA hairpin (hpRNAs), whereas siRNAs are 20–24 nt sequences derived from lengthy dsRNA precursors [3,4]. RNAi regulates a variety of biological processes, including plant immunity [5], and siRNAs and miRNAs have been identified as key factors in plant defense against viruses, bacteria, and fungi [6,7,8,9]. As shown in Figure 1, the silencing process starts with the binding of a host’s ribonuclease-III called Dicer (DICER) to long dsRNAs or hpRNA and their cleavage into siRNAs of 21–25 nt in length [10,11]. DICER has a helicase domain, a Piwi/Argonaute/Zwille (PAZ) motif, a dsRNA binding domain at the N-terminus, and two RNase III motifs at the C-terminus. DICER-generated siRNAs are subsequently integrated into the RNA-induced silencing complex (RISC). This multicomponent protein complex contains an Argonaute protein (AGO) with an sRNA-binding domain and endo-nucleolytic activity for RNA cleavage, which is triggered by the ATP-dependent unwinding of the siRNA duplex [12] (Figure 1). The passenger strand is degraded, and the guide strand binds to the target mRNA sequence and stimulates endonucleolytic cleavage or inhibits translation once the siRNA is integrated into RISC [13,14]. The existence of an RNA-dependent RNA polymerase (RdRP), which can interact with the RISC complex and create new dsRNA based on the partially degraded target template utilizing the hybridized siRNA strands as primers, is assumed to be the cause of this effect (Figure 1). Then, the DICER enzyme acts on the synthetized dsRNA to make additional siRNAs (secondary siRNAs). Once a dsRNA has been delivered into a cell, its influence can last throughout development; moreover, dsRNAs can be exported to neighboring cells, spreading the knockout gene effect throughout the organism [15]. There is growing evidence that sRNAs are mobilized in bidirectional interactions between plants and their pathogens, laying the groundwork for cross-kingdom RNAi (ck-RNAi) as a plant defensive mechanism [6,9,16].

Several studies have been conducted to investigate siRNA uptake in various organisms, and two primary mechanisms for host-derived RNA absorption have been proposed. One mechanism is siRNA absorption via plant-derived extracellular vesicles (EVs), which is based on the occurrence of exosome-like vesicles in plants that can carry bioactive compounds such as sRNAs to animal cells [17,18,19] (Figure 1). For example, in the fungal pathogen *Sclerotinia sclerotiorum*, using live cell images, it was concluded that the uptake of dsRNA occurs via clathrin-mediated endocytosis [20]. The other proposed mechanism occurs via plasma membrane-located transporters (Figure 1). This mechanism was supported by a study with the transmembrane protein SID-1, expressed in *Drosophila* S2 cells, which enabled passive dsRNA uptake from a culture medium [21]. Later, the lysosome transmembrane protein SIDT2 was identified in mammals and was shown to be involved in RNA uptake and subsequent degradation in this organelle [22].

This process of RNA trafficking from plant host cells to interacting pathogens has also been described in a variety of plant pathogenic fungi and oomycetes, such as *Botrytis cinerea*, *Cochliobolus sativus*, *Fusarium graminearum*, *Plasmopara viticola*, *Podosphaera xanthii*, *Sclerotinia sclerotiorum* and *Venturia inaequalis* [23,24,25,26,27,28,29,30]. Currently, the mechanisms of the transfer of sRNAs from plants to pathogenic fungi are unknown; however, the discovery that these eukaryotic pathogens are inhibited by sRNAs targeting their essential and/or pathogenicity genes has raised the possibility that plants could be protected by a new generation of environmentally friendly RNA-based fungicides that can be extremely specific and easily adaptable to control multiple diseases at the same time [31]. In this review, we will address the impact of RNAi technology on the research and management of two important groups of plant fungal pathogens, powdery mildew and rust fungi. First, we provide a brief description of the biological peculiarities of these fungi. Then, we describe how RNAi approaches have contributed to the analysis of gene function and have opened up new strategies for the management of powdery mildew and rust diseases, which are among the most damaging plant diseases.

## 2. Powdery Mildew and Rust Fungi

Obligate biotrophic fungi are a group of the most damaging plant pathogens, incurring massive economic losses and jeopardizing global food security. Powdery mildew and rust fungi infect more than 10,000 plant species, including many agronomically important crops, such as cereals, grapevines, many vegetables, and fruits, as well as ornamental and forest plants [32]. Their complete dependence on the host to feed, grow, and reproduce significantly complicates their manipulation under laboratory conditions, hindering research on their lifestyle and pathogenicity mechanisms at the molecular level [33].

Powdery mildew fungi are phytopathogenic ascomycetes belonging to the *Erysiphaceae* family, order *Erysiphales*, which includes 900 species and more than 80 genera. They cause damage in a wide range of angiosperm hosts, including both monocotyledons and dicotyledons plants. The fungal pathogens belonging to this group are easily identified by their symptoms, including the presence of powdery white patches on leaf surfaces, petioles, stems, blooms, and even fruits [34,35] (Figure 2A). In general, powdery mildew fungi exhibit both asexual and sexual life cycles (Figure 2B). The latter is highly uncommon for some species and only occurs under suitable environmental and nutritional conditions [36]. The asexual cycle starts after a conidium settles on a susceptible host plant. After its germination, it forms a small primary germ tube that elongates to become an appressorium (Figure 2B), which is in charge of penetrating the cuticle [37]. Subsequently, a hyphal peg will penetrate the epidermal cell creating a primary haustorium [38]. Upon effective infection, the main hyphae will branch and generate secondary hyphae and secondary haustoria. Later, conidiophores will emerge vertically from hyphae, generating a varying number of conidia or asexual spores depending on the species [36,39,40,41,42] (Figure 2B). This epiphytic fungal growth causes typical powdery mildew disease signs. In the event of sexual reproduction, two mating opposite hyphae need to be in contact to create a fruiting body termed chasmothecium, which holds one or more ascus containing the ascospores or sexual spores (Figure 2B) [41,42]. Although the exact infection structures developed by ascospores have not yet been determined, it is assumed that they are similar to those developed by conidia [35,43].

On the other hand, rust fungi comprise two orders, *Uredinales* and *Pucciniales*, in the widely varied phylum of Basidiomycota formed by mushrooms and bracket fungi. Rust fungi are divided into 14 families and 166 genera. Most species are found in the genera *Puccinia* and *Uromyces*, which have approximately 5000 and 1500 taxon names listed in Index Fungorum 2013, respectively [44]. Like powdery mildews, rusts are obligate biotrophic and pathogenic fungi that live on vascular plants ranging from ferns to monocots and gymnosperms to angiosperms (Figure 3A) [45,46,47,48]. Rust fungi have a typical macrocyclic-heteroecious life cycle where meiosis occurs in short-lived basidia formed by germinating teliospores (Figure 3B). Haploid basidiospores infect the aecial host and develop protoaecia and pycnia, among other fungal structures (Figure 3(B1,B2)). Pycnial nectar droplets create haploid pycniospores and receptive hyphae, where fertilization can take place between spores, and receptive hyphae of suitable mating types (Figure 3(B2,B3)). Following plasmogamy, dikaryotic aecia differentiate inside the host, and aeciospores are liberated and distributed by the wind (Figure 3(B3)). Aeciospores infect the telial host, causing the production of uredinia and urediniospores, which is followed by recurrent cycles of vegetative development on the telial host for several weeks or months, usually throughout the summer. Uredinia develops into telia in early fall, going through an overwintering phase during which karyogamy occurs, resulting in diploid dormant teliospores Figure 3(B4,B5) [45,48,49].

Both powdery mildew and rust fungi share a special structure of parasitism developed inside plant cells termed the haustorium. This specialized cell has been shown to deploy effectors, which are secreted proteins translocated into the plant cell, responsible for promoting the manipulation of the plant’s immune system and orchestrating the reprogramming of gene expression from the infected tissue to maintain fungal growth and development upon a successful infection [50,51]. The haustorium is also involved in the uptake of nutrients such as carbohydrates and amino acids and potentially water from the host via ion pumps present in the plasma membrane [52]. In addition, its ability to take genetic material such as dsRNA or siRNA makes it a key element in the development of methods of genetic transformation for biotrophic fungi, opening a world of possibilities that will allow many processes and functions to be studied in depth in the future [53,54,55].

## 3. RNAi Tools for Gene Function Analysis of Obligate Biotrophic Fungi

A major limitation of molecular studies in powdery mildew and rust fungi is their genetic intractability, probably due to their lifestyle as obligate biotrophs. To date, a number of transformation methods for filamentous fungi have been developed [56]. Some have been tested in powdery mildews and rusts, but unfortunately, the transformation is unstable, and the number of transformants is very low [55,57]. To mitigate this situation in part, a number of RNAi approaches have been developed for gene function analysis of these obligate biotrophic fungi, such as virus-induced gene silencing (VIGS), host-induced gene silencing (HIGS), Agrobacterium tumefaciens-mediated HIGS (ATM-HIGS) and the direct application of dsRNA, which are described below (Figure 4):

### 3.1. Virus-Induced Gene Silencing (VIGS)

VIGS is a term used to describe a tool that employs recombinant viruses to induce gene silencing in response to genetically manipulated RNA viral vectors [58]; Figure 4A. This technique was described for the first time in *Nicotiana benthamiana*, where cDNA fragments of the *N. benthamiana* phytoene desaturase (*PDS*), a gene involved in the carotenoid biosynthesis pathway, were inserted into a hybrid viral vector composed of sequences from the tobacco and tomato mosaic viruses (TMV and ToMV). These viral constructs resulted in an inhibition of carotenoid synthesis downstream of phytoene and the rapid destruction of chlorophyll by photooxidation, resulting in a white leaf phenotype in plants [59]. Since then, this approach has become a powerful silencing tool for species where stable transformants are difficult to obtain. The most popular vector for VIGS used in monocotyledons and dicotyledons is barley stripe mosaic virus (BSMV), comprising the tripartite genome RNAa, RNAß and RNAγ [60]. RNAα encodes the replicase protein (αa), RNAβ encodes a coat protein (βa) and three movement proteins (βb, βc, and βd), and RNAγ encodes the polymerase (γa) component of the replicase and the site where the fragments of target fungal genes are inserted (usually in the antisense orientation) directly downstream of the stop codon of ORFγb. For each experimental scenario, the modified RNAγ is mixed with RNAα and RNAβ and inoculated into host plants [61]. To our knowledge, the first unique evidence of the use of VIGS for powdery mildew gene silencing was described by Nowara and colleagues, who silenced two *B. graminis* 1-3 β-glucosyltransferase (*BgGTF1* and *BgGTF2*) genes using a BSMV-VIGS system and reported a reduction in fungal growth on wheat (Table 1). In rust fungi, a VIGS approach was developed to identify gene function in *Puccinia striiformis* f. sp. *tritici*. The system was used to determine the *Puccinia*-specific gene silencing signal from the plant to the pathogen suppressing fungal gene expression. For this proposal, five predicted secreted proteins of *P. striiformis* (PSTha12J12, PSTha5A23, PSTha12H2, PSTha2A5, PSTha9F18), one chitinase predicted protein (PSTha5A1) and a homologue to *Uromyces fabae* hexose transporter (PSTha12O3) were silenced (Table 1). While reductions in rust development or sporulation were not observed for any of the genes tested, the results showed that VIGS could be used for functional gene analyses in rust fungi [62]. Then, new rust fungal targets were studied, such as *PsCNA1* and *PsCNB1*, which are involved in the calcineurin signaling pathway that appears to be related to rust morphogenetic haustorium differentiation during the early stage of infection and production of uredospores [63] (Table 1). Another study targeted the protein kinase gene *PsSRPKL*, resulting in not only a reduction in fungal growth but also an increase in reactive oxygen species (ROS) accumulation in the host [64] (Table 1). In the same species, the transient silencing of the genes encoding the adenine nucleotide translocase PsANT [65], the Zn-only superoxide dismutase PsSOD1 [66], the small GTP-binding protein PsRan [67], the MAPK kinase PsFUZ7 [68], the transcription factor PstSTE12 [69], the PKA catalytic subunit PsCPK1 [70], the MADX-box transcription factor PstMCM1-1 [71], the MAP kinase kinase kinase PsKPP4 [72], the secreted protein Pst_8713 [73], and the effector protein PstGSRE1 [74] resulted in a substantial reduction of fungal growth, diminution in the spread of the hyphae and an impaired pathogenesis capacity, with some of the genes appearing to have a role in suppressing plant immunity or cell death (Table 1). Similarly, the use of the VIGS system produced a decrease in the expression of *P. triticina* genes, including mitogen-activated protein kinase 1 (*PtMAPK1*), cyclophilin (*PtCYC1*) and calcineurin B (*PtCNB*), which are involved in the establishment of disease in host plants, reducing disease symptoms and fungal growth [75] (Table 1). Following the same methodology, other studies in *P. graminis* f. sp. *tritici* determined that the transient silencing of genes such as the putative tryptophan mono-oxygenase *Pgt-IaaM* or genes involved in different functions such as glycosylation, sugar metabolism, transport, thiazole biosynthesis, secreted protein or unknown function (PGTG_01136, PGTG_01215; PGTG_03478, PGTG_10731, PGTG_12890, PGTG_01304, PGTG_16914, PGTG_03590, PGTG_14350) reduces not only fungal growth but also the size of urediniospores [76,77] (Table 1).

### 3.2. Host Induced Gene Silencing (HIGS)

The HIGS strategy results in the silencing of a pathogen-specific gene through *in planta* expression of dsRNA homologous to the pathogen’s target gene of interest [15]. Micro-bombardment is one of the methods used for delivering siRNA molecules into plant cells for HIGS. The high-velocity particles penetrate the cell wall and membrane, releasing the siRNA molecules into the cell cytoplasm. Once inside the cell, siRNAs can target specific mRNA molecules of a pathogenic organism, leading to their degradation or translational repression and hence silencing the expression of the pathogen’s genes [74,78]. Control of pathogen growth occurs due to RNAi-mediated silencing of a target gene related to pathogen growth and/or development, including pathogen-related structures to pathogenesis or by silencing those that are negative regulators of the host defence. Its success is based on the ability of the powdery mildew and rust fungi to take up, presumably through the haustorium, hpRNA or other RNAi molecules produced by plant cells after transformation with the silencing constructs (Figure 4B) [75,79].

The use of the HIGS method has been mainly described for powdery mildews. It has been more than a decade since Nowara and collaborators developed the approach based on HIGS by the exchange ability of siRNA molecules between cereal cells and the obligate biotrophic fungus *B. graminis* f. sp. *hordei* through a gene silencing method using dsRNA targeting the avirulence gene *Avra10*. The results of this assay showed a fungal growth reduction in the absence of the resistance gene *Mla10* (Table 2). These results also suggested that these fungal genes play a role in haustorium formation and elongation of secondary hyphae [53]. Subsequently, the HIGS technique was applied to study many secreted proteins in the *Blumeria* species, such as the silencing of eight effector candidates obtaining a significant decrease in pathogen development [54] (Table 2). Another research study analyzed the candidate-secreted effector protein (CSEP; CSEP0055), and the results showed a reduction in the formation of haustoria [80] (Table 2). Later, HIGS in other CSEPs, such as CSEP0105 and CSEP0162 [81] or CSEP0027 [82], which stabilize several intracellular factors, including defense-related signaling components or CSEP007, CSEP0025, CSEP0128, CSEP0247, CSEP0345, CSEP0420, CSEP0422, CSEP0081, and CSEP0254, which are involved in early fungal aggressiveness [83,84]. As well as CSEP0139 and CSEP0182, which suppress host cell death, also resulted in a significant reduction in fungal penetration and haustoria formation rate [85] (Table 2). Although HIGS was useful in demonstrating the role of several candidate genes, the function of many others remains unknown, leaving the door open for further research.

### 3.3. Agrobacterium tumefaciens-Mediated Host-Induced Gene Silencing (ATM-HIGS)

Although the so-called HIGS system has allowed the individual study of various fungal CSEPs, the method, which requires particle micro-bombardment for the transformation of plant cells, has certain disadvantages, such as low-frequency success in transformation and integration and randomness of the intracellular target (cytoplasm, nucleus, vacuole, plastid, etc.), among others [86,87]. With the finding that the virulence mechanism of *Agrobacterium tumefaciens* leads to tumor formation, plant biotechnologists adapted the HIGS system as a new tool for the transient transformation of plants. RNAi-based gene silencing mediated by *A. tumefaciens* (ATM-HIGS) uses a vector consisting of a tumor-inducing plasmid (Ti plasmid) in which the oncogenes responsible for the formation of tumors of the region known as T-DNA (transferred DNA) are replaced by the RNAi machinery for the formation of dsRNA of the target gene [88]. Thus, the *Agrobacterium* system produces a transient transformation in plant cells due to the delivery of RNA-silencing molecules into leaf cells or other plant tissues [89,90,91] (Figure 4C).

Panwar and collaborators performed the first *Agrobacterium*-mediated gene silencing a says to demonstrate its silencing ability in biotrophic fungi using genes encoding mitogen-activated protein kinase 1 (*PtMAPK1*), cyclophilin (*PtCYC1*) and calcineurin B (*PtCNB*) from the rust fungus *P. triticina* [75] (Table 3). Since then, ATM-HIGS has been used successfully for gene function analysis of candidate effectors of the powdery mildew fungus *P. xanthii*, such as phospholipid-binding protein (PEC019), α-mannosidase (PEC032), cellulose-binding protein (PEC054), effectors with chitinase activity (EWCAs), lytic polysaccharide mono-oxygenase (*PHEC27213*), chitin deacetylases (*PxCDA1* and *PxCDA2*) and other proteins with unknown functions [55,92,93,94] (Table 3). As most of these genes contribute to fungal virulence, their knockdown resulted in a substantial restriction in fungal growth and a significant increase in the plant’s immune response (Table 3).

### 3.4. Direct Application of dsRNA

In recent years, the use of exogenous dsRNA, sRNAs and hpRNAs has gained prominence as a new alternative that could be regarded as more sustainable, applicable and easily introduced into the host compared to the rest of the tools already discussed [95,96] (Figure 1 and Figure 4D). In phytopathogenic fungi, there are several studies that corroborate the efficacy of the use of exogenous dsRNA molecules for several gene function analyses [23,26,97,98]. One of the examples was performed by McLoughlin and collaborators using dsRNAs directed at genes related to transcription or host colonization of the fungi *Sclerotinia sclerotiorum* and *B. cinerea*. The results of this study showed a significant decrease in fungal infection and a reduction in disease symptoms [24].

The efficacy of this technique was also evaluated by infiltration of dsRNAs targeting CSEPs in the obligate biotrophic fungus *Erysiphe pisi* (*EpCSEP001*, *EpCSEP009* and *EpCSP083*), showing a significant reduction in disease symptoms and demonstrating the involvement of these genes in the pathogenesis of pea plants [99] (Table 4). Similarly, a functional analysis of several conserved and non-annotated proteins (CNAPs) in *P. xanthii*, presumably involved in essential functions such as respiration (CNAP8878, CNAP9066, CNAP10905 and CNAP30520), glycosylation (CNAP1048) and efflux transport (CNAP948), showed a potential reduction in cucurbit powdery mildew disease after the infiltration of dsRNA targeting these genes [29] (Table 4). Recently, this approach has also been tried on Asian soybean rust targeting chitin synthase (CHS) genes and resulted in a large reduction in fungal lesion formation [100] (Table 4).

## 4. Control of Powdery Mildew and Rust Diseases by RNAi Technology

New insights into the ability of RNA molecules to move across cellular boundaries between hosts and pathogens and their ability to specifically repress essential genes of various pathogens have led to the development of novel disease management strategies [101]. The RNAi strategies developed to control powdery mildew and rust diseases are described below.

### 4.1. Transgenic Plants Expressing RNAi Constructs

In the last decade, several studies have proposed the use of stable HIGS in plants to confer disease resistance to fungal pathogens [53,102,103,104]. Based on experimental validation from transient HIGS assays in barley, transgenic barley plants that expressed antifungal RNAi constructs targeting the *B. graminis* f. sp. *hordei GTF1* gene, which encodes a 1,3-β-glucanosyltransferase belonging to the penetration-associated *cap20* regulon, were tested [53,105] (Table 5). Three T1 lines showed a significant reduction in *B. graminis* disease symptoms when a transgenic control line, which had lost the hairpin RNAi cassette, was as susceptible to powdery mildew fungus as the non-transgenic control plants. To date, fungal pathogenesis-related genes and housekeeping genes have been the primary targets for stable HIGS; however, the increased interest in effector studies in powdery mildew encouraged the question of whether their silencing would be similar to that obtained with essential fungal genes. To answer this question, Schaefer and colleagues (2020) focused on *B. graminis* f. sp. *tritici* effectors *SvrPm3^a1/f1^*, *Bgt-Bcg-6*, and *Bgt_Bcg-7*, one of the largest classes of candidate effectors in the *Blumeria* genomes, belonging to the RNase-like class [106,107]. In this study, stable HIGS of the three *B. graminis tritici* effectors resulted in a quantitative gain of powdery mildew resistance in wheat. These resistance events could impair haustorium formation on seedlings and restrict fungal growth on leaves (Table 5).

Similarly, the expression of RNAi constructs targeting the MAPK kinase gene *PsFUZ7* in transgenic wheat plants conferred strong and genetically stable resistance to the devastating stripe rust fungus *P. striiformis* f. sp. *tritici* [68] (Table 5). In this study, two independent transgenic lines, which were highly effective in restricting the spread of *P. striiformis*, were selected in the T3 generations and examined to verify whether this phenotype was caused by the production of siRNAs corresponding to the targeted *PsFUZ7* sequences. Gene expression and biomass analyses showed that both transgenic lines exhibited a significant reduction in *PsFUZ7* transcripts and fungal biomass. Moreover, histological observations revealed differential hyphal growth in transgenic lines carrying *PsFUZ7* RNAi constructs compared to the control, supporting the important role of *PsFUZ7* in *P. striiformis* virulence by regulating mycelial growth and development (Table 5). Another excellent target to generate durable genetic resistance against wheat stripe rust was *PsCPK1*, a protein kinase A (PKA) catalytic subunit gene from *P. striiformis* that is highly conserved in fungi and is involved in virulence, morphogenesis, and development [108,109]. The hairpin silencing constructs of *PsCPK1* expressed in wheat plants were sufficient to suppress disease development of *Pst* in T4 generation lines, indicating durable resistance at the genetic level against rust infection [70] (Table 5). Generally, the T3 generation is considered the initial true transgenic line in hexaploid wheat [110]; therefore, it is significant that transgenic resistance to *P. striiformis* was identified up to the fourth generation [70]. In the same way, the expression of RNAi constructs targeting the *Pst_4* and *Pst_5* rust effectors resulted in weaker hyphal development and larger H_2_O_2_ accumulation in transgenic plants compared with the non-transgenic control plants against *P. striiformis* [111] (Table 5). Later, the development of transgenic wheat plants that stably expressed RNAi constructs of pathogenicity target genes of *P. triticina* resulted in effective resistance against wheat leaf rust (WLR) disease [112] (Table 5). In particular, the engineered resistance trait was heritable and stable in the T2 generation, and the suppression of WLR development was correlated with the presence of siRNA molecules specific to the fungal *PtMAPK1* and *PtCYC1* genes [75].

Although the results described previously were promising, the application of HIGS by transgenic expression may be restricted by several factors: the difficulty or impossibility of transforming several crop species, the public concern about the biosafety of genetically modified crops, and the instability of artificial RNAi constructs [111]. These factors could complicate the generation of genetically modified crops [113]; consequently, a plant disease management strategy that does not rely on transgenic approaches is highly desired for environmentally sustainable agriculture.

**Table 5 ijms-24-09082-t005:** RNAi transgenic plants for the control of powdery mildew and rust diseases.

Plant Host	Cultivar	Pathogen	Target Gene	Gene Function	Effects	References
*H. vulgare*	Golden Promise	*Blumeria graminis*	*BgGTF1*	1,3-β-glucanosyltransferase 1	Reduced manifestation of powdery mildew symptoms	[53]
*T. aestivum*	Bobwhite	*B. graminis* f. sp. *tritici*	*SvrPm3^a1/f1^*	RNase-like effector	Enhanced resistance to powdery mildew	[9]
			*Bgt-Bcg-6*			
			*Bgt-Bcg-7*			
	Xinong1376	*Puccinia striiformis* f. sp. *tritici*	*PsFUZ7*	MAPK kinase	Enhanced resistance to rust	[68]
			*PKA*	Protein kinase A	Enhanced resistance to rust	[70]
			*PsCPK1*	Catalytic subunit		
	Fielder		*Pst_4*	Effector	Enhanced resistance to rust	[111]
			*Pst_5*			
	Fielder	*Puccinia triticina*	*PtMAPK1*	MAP kinase	Reduction of wheat leaf rust disease symptoms	[111]
			*PtCYC1*	Cyclophilin		

### 4.2. Spray-Induced Gene Silencing (SIGS)

To circumvent transgenic approaches, an innovative new strategy designated spray-induced gene silencing (SIGS) has been recently developed, which induces the silencing of pathogen target genes without the need to develop stably transformed plants and available transformation protocols. This RNAi-based technology allows the inhibition of pathogens and disease development by topical application of siRNA or dsRNA molecules onto plants to silence essential plant pathogen genes [101] (Figure 5). To date, SIGS has been demonstrated to be effective in controlling a wide range of plant pathogenic fungi [24,26,114].

Recently, the potential of SIGS has also been tested against powdery mildew and rust diseases. The first study of suppressing cucurbit powdery mildew through SIGS was reported by Ruiz-Jiménez et al. (2021) [29]. Spray application of dsRNAs targeting three *P. xanthii* genes essential for fungal development induced high levels of disease control. In all cases, disease severity was reduced by approximately 80% to 90% compared to water-treated melon leaves [29] (Table 6). Furthermore, in this study, the efficacy of SIGS was tested using various doses of dsRNA, and the results indicated that such dsRNAs remained functional at concentrations as low as 5 μg/mL. However, higher concentrations of dsRNA seemed to provide higher disease control, as previously demonstrated [23]. In rust fungi, Hu et al. (2020) studied the efficacy of silencing the Asian soybean rust fungus *Phakopsora pachyrhizi* through SIGS. In this study, direct spraying of dsRNAs targeting genes encoding an acetyl-CoA acetyltransferase (ATC), a 40S ribosomal protein S16 (RP_S16), and a glycine cleavage system H protein (GCS_H) onto soybean leaves was able to reduce the number of pustules per cm^2^ of leaf, fungal biomass, and endogenous target gene expression by at least 68% compared to control soybean leaves sprayed with water [115] (Table 6). In fact, SIGS targeting *P. pachyrhizi* chitin synthase (*CHS*) genes resulted in a reduction in soybean rust lesions and appressoria formation by more than 40% [100]. On the other hand, exogenous application of dsRNA targeting essential genes of *Austropuccinia psidii* (the cause of myrtle rust) significantly reduced infection in whole plants [116] (Table 6).

Although further studies are needed, these early successes of SIGS approaches support the idea that RNAi technology could be used to combat powdery mildew and rust diseases in a sustainable and environmentally friendly manner. This strategy does not require the development and approval of genetically engineered technologies for each crop species. It does not limit its application to a single gene or pathogen, as it is possible to target multiple essential genes of different pathogens simultaneously [117,118]. This new class of RNA-based fungicides could offer many advantages over conventional chemical treatment. However, under field conditions, the effectiveness of dsRNAs acting as fungicides may be uncertain due to the instability of RNA molecules in the environment. For this reason, current research efforts are focusing on the use of nanoparticles as carriers to deliver biologically active dsRNA, expanding the duration of their silencing effect in field conditions [119]. Currently, the nanoparticles developed for the application of these oligonucleotides in plants include inorganic and organic nanoparticles. Among the inorganic materials, those of layered double hydroxides (LDHs), carbon dots (CDs), carbon quantum dots (CQDs) or gold nanoparticles, among others, stand out. LDH nanoparticles have been used to prolong dsRNA activity and protect against viruses [120], insects [121] and fungal pathogens [122,123]. Regarding CDs, in a recent publication, dsRNAs coated with CDs were delivered to cucumber plants, leading to promising results in the control of cucurbits viruses [124]. Regarding CQDs, Kostov and colleagues (2022) found that the mixture of CQDs with dsRNA increased RNAi efficiency by causing a significant reduction in the transcript levels of the target gene in developing sporangia [125]. On the other hand, the use of organic nanoparticles as carriers has also shown interesting results. To mimic the natural mechanisms by which plants deliver their own siRNAs to pathogens, dsRNAs packaged in liposomes or in extracellular synthetic phospholipid bilayers have been used [126]. Finally, the emergence of DNA nanotechnology has also provided a promising and highly tunable platform with which to design, synthesize and utilize DNA nanostructures to deliver cargoes (drug, DNA, RNA and protein) to bypass the plant cell wall for gene silencing applications passively. It was recently demonstrated that DNA nanostructures could be used as cargo carriers for direct siRNA delivery and gene silencing in mature tobacco plants [127].

**Table 6 ijms-24-09082-t006:** Control of powdery mildew and rust diseases by Spray-induced gene silencing.

Plant Host	Cultivar	Pathogen	Target Gene	Possible Gene Function	RNA Amount	RNA Application	Effects	References
*C. melo*	cv. Rochet	*Podosphaera xanthii*	*PxCNAP1048*	Glycosylation	5–30 μg/mL	Leaves were spray-inoculated with 10^4^ conidia/mL after dsRNA application	Effective management of PM disease	[29]
			*PxCNAP10905*	Respiration				
			*PxCNAP30520*					
*G. max*	cv. Enrei	*Phakopsora* *pachyrhizi*	*ATC*	Acetyl-CoA acyltransferase	20 μg/mL	Leaves were spray-inoculated with 10^5^ uredinia/mL after dsRNA application	Effective management of Asian soybean rust (ASR) disease	[115]
			*RP_S16*	40S ribosomal protein S16				
			*GCS_H*	Glycine cleavage system H protein				
			*CHS*	Chitin synthase	10 ng/mL	Leaves were drop-inoculated with 10^5^ uredinia/mL and dsRNA simultaneously	Effective management of Asian soybean rust (ASR) disease	[100]
*Syzygium jambos*	-	*Austropuccinia psidii*	*β-TUB*	β-tubulin	100 ng/μL	Young, emerging leaves were inoculated with 1 mL of dsRNA solutions	Reduction in fungal growth and in the number of urediniospores	[116]
			*EF1-a*	Translation elongation factor 1ɑ				
			*ATC*	Acetyl-CoA transferase				
			*CYP450*	Cytochrome P450				
			*MAPK*	Mitogen-activated protein kinase				
			*GCS-H*	Glycine cleavage system H				
			*28S rRNA*	28S ribosomal RNA				
			*HAUS01215*	Haustoria target				

## 5. Conclusions and Future Prospects

To date, chemical control has been the most effective disease management strategy against powdery mildew and rust diseases. However, the increase in public concern about the use of chemicals and the emergence of fungicide-resistant isolates have resulted in a situation where novel alternative approaches to fungicide applications are urgently needed. The discovery of cross-kingdom RNAi has provided not only new approaches for gene function studies but also a new environmentally friendly and non-transgenic tool for the management of fungal plant diseases, including those caused by powdery mildew and rust fungi. The use of RNAi-based fungicides via SIGS can circumvent the problems associated with transgenic crops through the direct application of siRNA or dsRNA molecules onto plants to provide protection against pathogens; however, the stability of these molecules under field conditions is considered a major concern that may limit the application of SIGS-based disease management strategies. Therefore, in future studies, the utilization of nanoparticles and other stabilizers could improve either dsRNA stability on plant tissues, which will reduce the application frequency for growers, or dsRNA uptake efficiency, which will reduce the amount of dsRNA needed. Another major problem to consider is the cost and low efficiency of dsRNA production. In this regard, new studies should be encouraged to develop cost-effective large-scale production of dsRNA for agricultural use to facilitate SIGS implementation. In addition, fundamental research is needed to unravel the mechanisms of sRNA uptake by these fungi, and this information may be crucial to understanding and optimizing RNAi-based gene silencing in these plant pathogens. RNAi-based fungicides and SIGS will soon be a major component of the arsenal of tools for managing powdery mildew and rust diseases, thus contributing to the advancement of the modern concept of organic and sustainable agriculture.

## Figures and Tables

**Figure 1 ijms-24-09082-f001:**
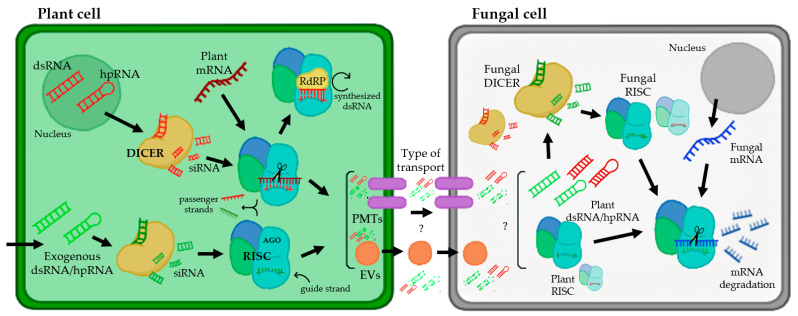
Interaction between a plant cell and fungal pathogen from the perspective of plant RNAi-mediated host-induced gene silencing. The arrows represent the flow of the gene silencing mechanism using RNAi. In the nucleus of a plant cell, dsRNA and hpRNA are produced as normal defense responses or from hairpin RNAs in transgenic RNAi plants (targeting a fungal gene). In addition, there are several biotechnology tools that allow the entry of exogenous dsRNAs or hpRNAs. These molecules of dsRNA and hpRNA can be processed by the DICER enzyme, creating siRNAs, which are integrated into the RNA-induced silencing complex (RISC), which contains an Argonaute protein (AGO), using them as templates for mRNA silencing. For the amplification of this silencing mechanism, there is an RNA-dependent RNA polymerase (RdRP) that can synthesize new dsRNAs using hybridized siRNA strands as primers. siRNAs produced in plant cells can be transported presumably by two types of transport (represented with question marks): via plant-derived extracellular vesicles (EVs) and plasma membrane-located transporters (PMTs). Inside the fungal cell, the mechanism of silencing works similarly to plant cells, producing the assembly of siRNAs with the RISC and inducing the silencing of specific fungal mRNAs.

**Figure 2 ijms-24-09082-f002:**
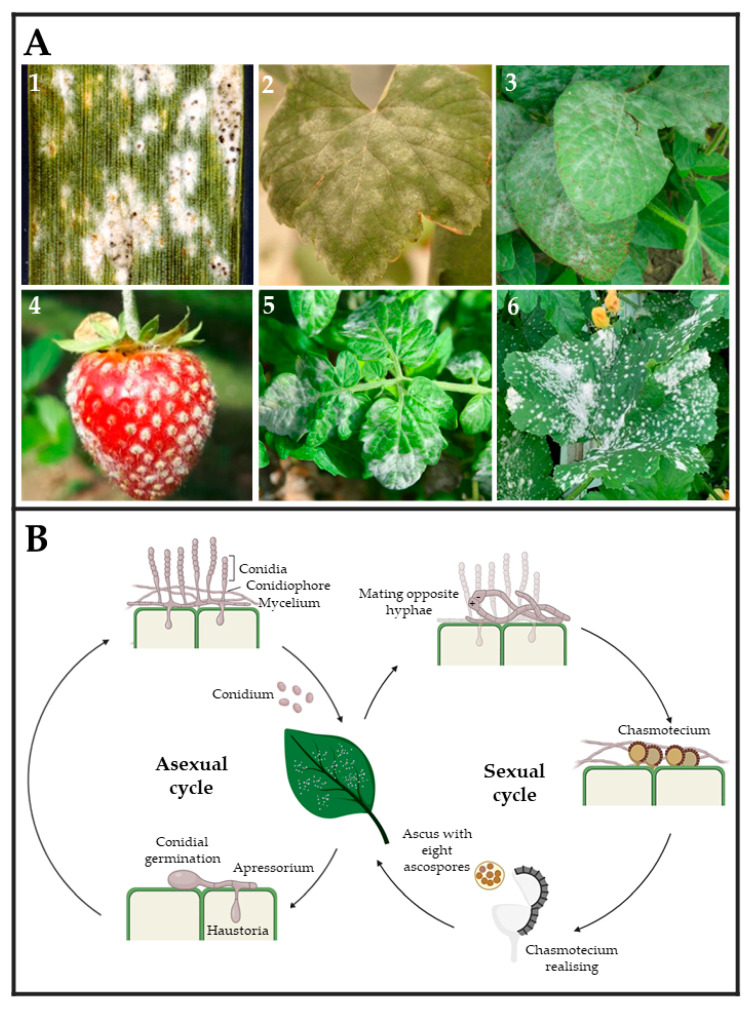
(**A**) Powdery mildew symptoms observed on leaves and fruits of several crops. (**1**) Wheat (*Triticum aestivum*) leaf, (**2**) wine grape (*Vitis vinifera*) leaf, (**3**) soybean (*Glycine max*) leaves, (**4**) strawberry (*Fragaria* sp.), (**5**) tomato (*Solanum lycopersicum*) leaves, and (**6**) melon (*Cucumis melo*) leaves infected by *Blumeria graminis*, *Erysiphe necator*, *Microsphaera diffusa*, *Podosphaera aphanis*, *Leveillula taurica*, and *Podosphaera xanthii*, respectively. Pictures (**1**)–(**6**) were taken by Clemson University—USDA CES, Yuan-Min Shen (National Taiwan University), Daren Mueller (Homemade, Bugwood.org (accessed on 28 March 2023)), University of Hertfordshire, Scot Nelson (Homemade flickr.com (accessed on 28 March 2023), and by the authors of this review, respectively. (**B**) The typical powdery mildew life cycle is divided into two types of reproduction. Asexual reproduction is carried out by the release of conidium spores, which develop haustoria capable of acquiring nutrients from plant cells and giving rise to hyphae and conidiophores. Sexual reproduction occurs when two hyphae from opposing mating types form a chasmothecium capable of releasing an ascus with eight ascospores.

**Figure 3 ijms-24-09082-f003:**
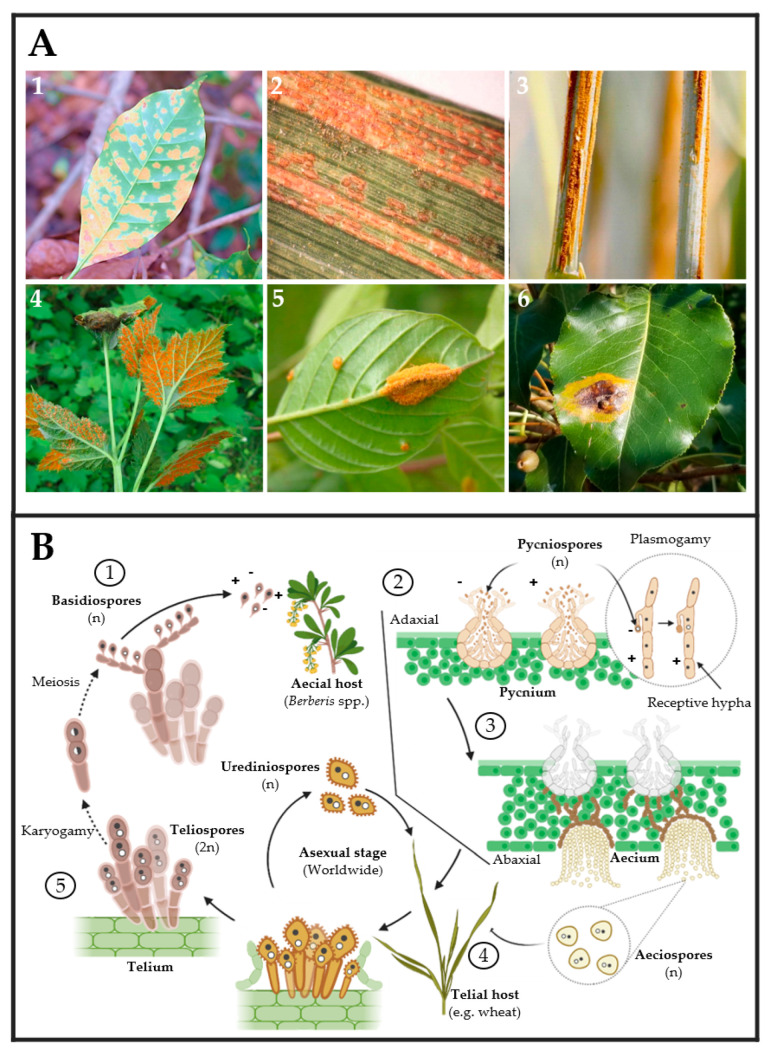
(**A**) Rust fungi symptoms observed on leaves of several crops: (**1**) coffee leaf (*Coffea arabica*), (**2**) barley leaf (*Hordeum vulgare*), (**3**) oat stem (*Avena sativa*), (**4**) black raspberry leaves (*Rubus occidentalis*), (**5**) glossy buckthorn (*Frangula alnus*), and (**6**) pear leaves (*Pyrus* spp.) infected by *Hemileia vastatrix*, *Puccinia striiformis*, *P. graminis* f. sp. *avenae*, *Arthuriomyces peckianus*, *Puccinia coronata* and *Gymnosporangium sabinae*, respectively. Pictures (**1**)–(**6**) were taken by Dr. Parthasarathy Seethapathy (Amrita School of Agricultural Sciences), Mary Burrows (Montana State University), Howard F. Schwartz (Colorado State University), Sandra Jensen (Cornell University), Milan Zubrik (Forest Research Institute—Slovakia; homemade, Bugwood.org (accessed on 28 March 2023)) and Sue Muller (Homemade MarylandBiodiversityProject.com (accessed on 28 March 2023)), respectively. (**B**) The typical rust fungal life cycle includes two types of hosts. First, aecial hosts are infected by haploid basidiospores ①, which generate pycnium as reproductive structures ②. Pycnium produce pycniospores with different polarities, which produce plasmogamy for generating haploid aeciospores ③. Second, aeciospores infect the telial host, in which urediniospores can be generated for asexual reproduction ④. In the telial host, teliospores are produced by karyogamy. Finally, basidiospores are produced by the meiosis of teliospores ⑤.

**Figure 4 ijms-24-09082-f004:**
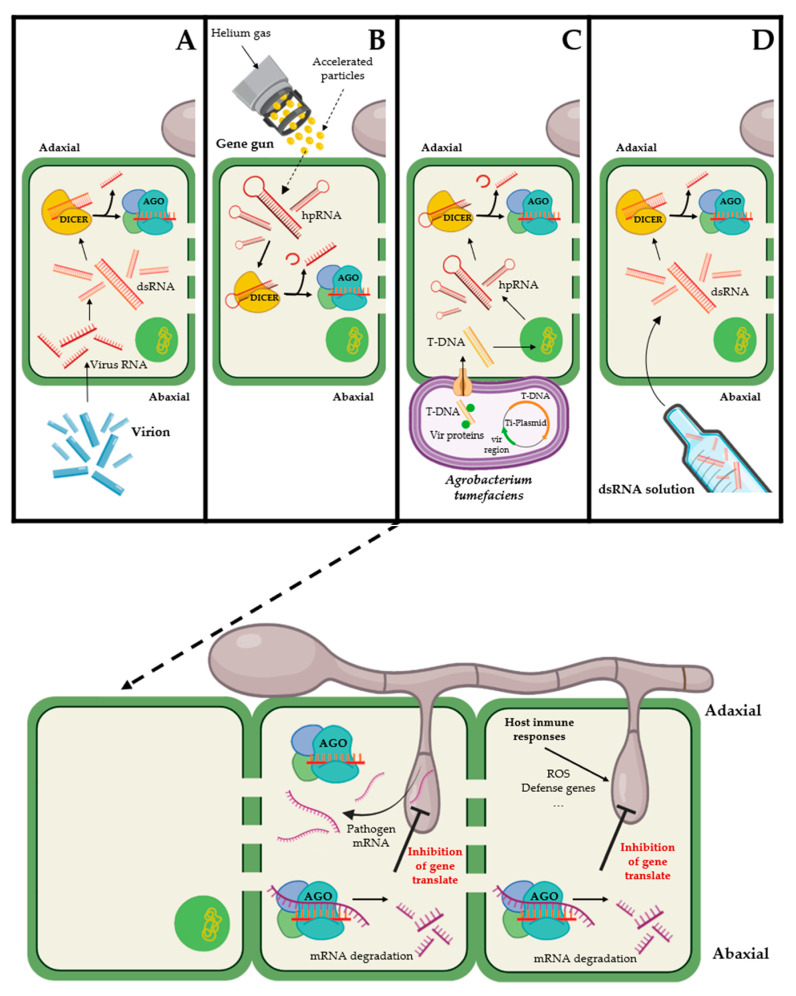
Models showing RNA interference tools for plant and fungal pathogens. (**A**) VIGS: Virus RNAs (α, β, γ) are inoculated into plant cells. Inside the host, barley stripe mosaic virus (BSMV) is assembled. dsRNAs specific for the pathogen mRNA target are produced by the virus machinery. dsRNA processed by the silencing machinery silences the expression of the specific mRNA. (**B**) HIGS with micro-bombardment; hpRNAs join accelerated particles and are used for transitory transformation of plant cells. Inside the plant, hpRNAs are cut by the Dicer enzyme, and the sRNA derivate activates the RISC complex, which is able to hybridize with mRNA targets, allowing mRNA degradation by the ARGONAUTE enzyme. (**C**) ATM-HIGS: *Agrobacterium tumefaciens* transformed with Ti plasmid with a specific sequence of the pathogen target gene produces a transitory transformation of the plant. The Ti plasmid has sequences of vir genes. These vir genes encode several vir proteins responsible for transporting T-DNA into host cells. Inside the plant, T-DNA is introduced into the genome of the cells, allowing the production of hpRNAs. These hpRNAs are used to silence pathogen mRNAs. (**D**) dsRNA infiltration: dsRNA produced in vitro is introduced directly into the plant. These dsRNAs are specific for the silencing of pathogen mRNA targets. The dsRNA is cut into sRNAs; sRNAs joined to the RISC complex can hybridize with specific mRNAs being degraded by the ARGONAUTE enzyme.

**Figure 5 ijms-24-09082-f005:**
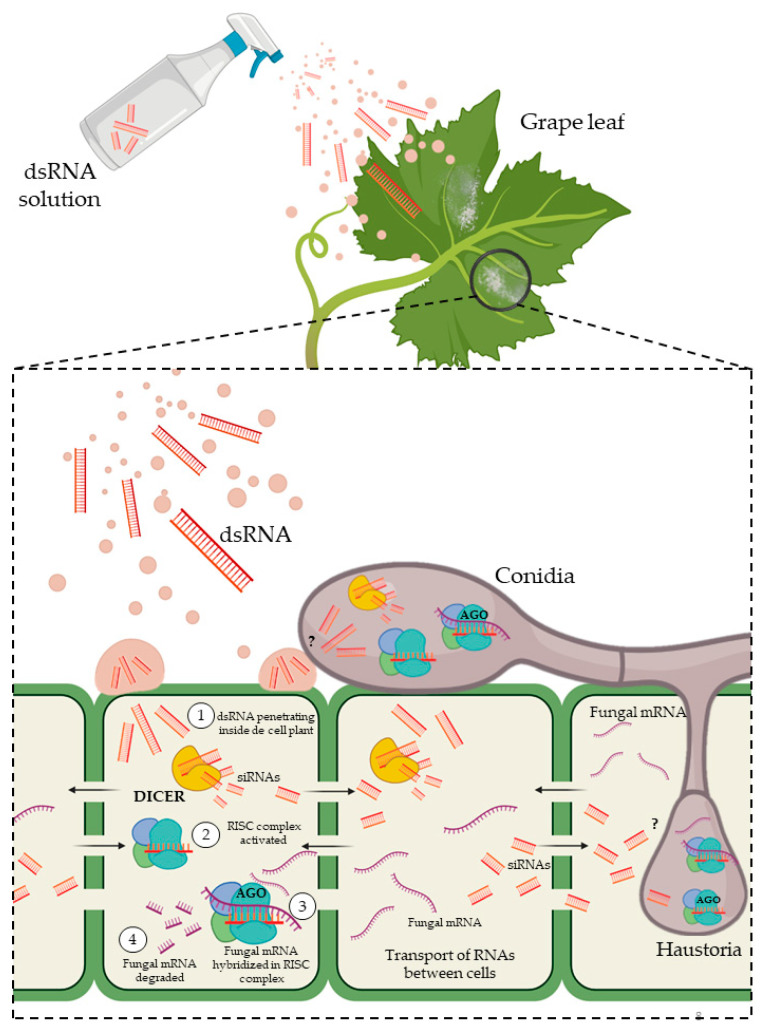
Molecular process scheme of SIGS assays for fungal control. The designed dsRNAs are sprayed onto the host plant leaf. The first way of uptake is by the plant ① with the subsequent activation of the DICER-RISC-mediated silencing system ②–③ degrading mRNA targets of the pathogen ④; however, some fungal structures, such as haustorium, can presumably uptake small molecules as siRNAs and activate the DICER-RISC complex inside the fungal cell as it is indicated in question marks.

**Table 1 ijms-24-09082-t001:** Virus-induced gene silencing method used for gene function analyses in rust and powdery mildew fungi.

Plant Host	Pathogen	Target Gene	Possible Gene Function	Application	Phenotype	References
*Hordeum vulgare*	*Blumeria graminis* f. sp. *hordei*	*GTF1*	Cell wall elongation and virulence factor	Virus inoculation by rubbing of barley first leaves	Reduction in haustorium formation	[53]
		*GTF2*				
*Triticum aestivum*	*Puccinia striiformis* f. sp. *tritici*	*PSTha12J12*	Predicted secreted protein	Virus inoculation by rubbing wheat leaves	Reduction in the expression patternsof the fungal genes	[62]
		*PSTha5A23*				
		*PSTha12H2*				
		*PSTha2A5*				
		*PSTha9F18*				
		*PSTha5A1*	Predicted to code for a chitinase protein			
		*PSTha12O3*	Homologous to Uromycesfabae hexose trans-porters			
		*PsCNA1*	Calcineurin A-like protein (CNA1)		Slower elongation of fungal hyphae and reduction of the production of uredospore	[63]
		*PsCNB1*	Calcineurin B-like protein (CNB1)			
		*PsSRPKL*	Protein kinase		Reduction of fungal growth and increases of ROS accumulation in host cells	[64]
		*PsANT*	Adenine nucleotide translocase		Attenuated the growth and development of virulent *Pst* at the early infection stage	[65]
		*PsSOD1*	Zn-only superoxide dismutase		Reduction of the virulence-associated with ROS accumulation	[66]
		*PsRan*	Small GTP-binding protein		Reduction of the number of haustoria and the length of infection hyphae	[67]
		*PsFUZ7*	MAPK kinase		Reduction of initial haustorium formation and elongation of secondary hyphae	[68]
		*PstSTE12*	Transcription factor		Reduction in the growth and spread of hyphae in *Pst* and weakened the virulence of *Pst* on wheat	[69]
*T. aestivum*	*P. striiformis* f. sp. *tritici*	*PsCPK1*	PKA catalytic subunit	Virus inoculation by rubbing wheat leaves	Reduction in the length of infection hyphae and disease phenotype	[70]
		*PstMCM1-1*	MADX-box transcription factor		Reduction of hyphal extension and haustorium formation	[71]
		*PsKPP4*	MAPK kinase		Reduction of haustorium number	[72]
		*Pst_8713*	Suppresses host defenses and contributes to the pathogenicity of *Pst*		Reduction of haustorium number	[73]
		*PstGSRE1*	Effector to defeat ROS-associated plant defense by modulating the subcellular compartment of a host immune regulator		Reduction in sporulation and in the fungi biomass	[74]
	*Puccinia triticina*	*PtCYC1*	Cyclophilin		Reduction in fungal growth and disease symptoms	[75]
		*PtMAPK1*	MAP kinase			
		*PtCNB*	Calcineurin regulatory subunit			
	*Puccinia graminis* f. sp. *tritici*	*Pgt-IaaM*	Tryptophan mono-oxygenase		Reduction in fungal growth and in the size of uredinia	[76]
		*PGTG_01136*	Predicted glycolytic enzyme		Reduction in fungal growth and in the size of uredinia	[77]
		*PGTG_01215*	Probably involved in cellular carbohydrate or sugar metabolism			
		*PGTG_03478*				
		*PGTG_14350*	Hypothetical secreted protein with homology to periplasmic components of prokaryotic transport systems			
		*PGTG_10731*	Hypothetical proteins			
		*PGTG_12890*				
		*PGTG_01304*	Protein involved in thiazole biosynthesis			
		*PGTG_16914*	Amino acid permease			
		*PGTG_03590*	Secreted protein			
		*Pgt-IaaM*	Tryptophan 2-monooxygenase enzyme			

**Table 2 ijms-24-09082-t002:** Host-induced gene silencing method used for gene function analyses in rust and powdery mildew fungi.

Plant Host	Pathogen	Target Gene	Possible Gene Function	Application	Phenotype	References
*H. vulgare*	*Blumeria graminis* f. sp. *hordei*	*Avra10*	Virulence effector	Microprojectile bombardment	Reduction in haustorium formation	[53]
		*BEC1054*	Ribonuclease-like protein		Reduction in haustorium formation	[54]
		*BEC1011*				
		*BEC1019*	Metalloprotease			
		*BEC1005*	Endo β1-3 glucanase			
		*CSEP0055*	Effector involved in secondary penetration events		Reduction in haustorium formation	[80]
		*CSEP0105*	Effector proteins		Reduction in haustorium formation	[81]
		*CSEP0162*				
		*CSEP0027*	Interacts with barley HvCAT1 to regulate the host immunity to promote fungal virulence		Reduction in haustoria formation	[82]
		*CSEP0007*	Possibly involved in penetration and/or establishment of primary haustoria		Reduction in haustoria formation	[83]
		*CSEP0025*				
		*CSEP0128*				
		*CSEP0247*				
		*CSEP0345*				
		*CSEP0420*				
		*CSEP0422*				
		*CSEP0081*	Candidate Secreted Effector Proteins	Microprojectile bombardment	Reduction in fungal growth and in haustorium formation	[84]
		*CSEP0254*				
		*CSEP0139*	Suppressed cell death triggered by BAX and NtMEK2DD		Reduction in haustoria formation	[85]
		*CSEP0182*				

**Table 3 ijms-24-09082-t003:** *Agrobacterium tumefaciens*-mediated host-induced gene silencing method used for gene function analyses in rust and powdery mildew fungi.

Plant Host	Pathogen	Target Gene	Possible Gene Function	Application	Phenotype	References
*T. aestivum*	*Puccinia triticina* *Puccinia graminis* and *Puccinia striiformis*	*PtCYC1*	Cyclophilin	Agroinfiltration through the abaxial surface of wheat seedling leaves	Reduction in fungal growth and sporulation	[75]
		*PtMAPK1*	MAP kinase			
		*PtCNB*	Calcineurin regulatory subunit			
*Cucumis melo*	*Podosphaera xanthii*	*PEC007*	Candidate effector	Agroinfiltration of melon cotyledons	Reduction of fungal growth and increasing of the production of hydrogen peroxide by host cells	[55]
		*PEC009*				
		*PEC034*				
		*PEC032*	α-Mannosidase			
		*PEC019*	Phospholipid-binding protein			
		*PEC054*	Cellulose-binding protein			
		*PEC1666*	Chitinase activity		Reduction of fungal growth and increasing of the production of hydrogen peroxide by host cells	[92]
		*PEC1961*				
		*PEC2158*				
		*PEC5191*				
		*PHEC27213*	Lytic polysaccharide mono-oxygenase (LPMO) prevents the activation of chitin-triggered immunity		Reduction of fungal growth and increasing production of hydrogen peroxide by host cells	[93]
		*PxCDA*	chitin deacetylase		Reduction of fungal growth and increasing production of hydrogen peroxide by host cells	[94]

**Table 4 ijms-24-09082-t004:** dsRNA-induced gene silencing method used for gene function analyses in rust and powdery mildew fungi.

Plant Host	Pathogen	Target Gene	Possible Gene Function	Application	Phenotype	References
*Pisum sativum*	*Erysiphe pisi*	*EpCSEP001*	Virulence factors	Second leaves of pea plants were infiltrated with 100 parts per million (ppm) EpCSEP/CSP-dsRNA	Reduction in disease symptoms	[99]
		*EpCSEP009*				
		*EpCSP083*				
*C. melo*	*Podosphaera xanthii*	*PxCNAP1048*	Presumably involved in glycosylation	Melon cotyledons were infiltrated with dsRNA solutions of the different target genes in concentrations between 100 and 1000 ng ml^−1^	Reduction in fungal growth and disease symptoms	[29]
		*PxCNAP10905*	Presumably involved in respiration			
		*PxCNAP30520*				
		*PxCNAP8878*				
		*PxCNAP9066*				
		*PxCNAP948*	Presumably involved in efflux transport			
		*PxTUB2*	Involved in β-tubulin synthesis			
		*PxCYP51*	Involved in ergosterol synthesis			
*Glycine max*	*Phakopsora pachyrhizi*	*CHS*	Involved in chitin synthases	Soybean plants were infiltrated with 10 ng ml^−1^ of ds*CHS*	Reduction in fungal growth and in the number of urediniospores	[100]

## Data Availability

Not applicable.

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
