# Peer review of "RNAi Technology: A New Path for the Research and Management of Obligate Biotrophic Phytopathogenic Fungi"

_ijms, 2023, doi:10.3390/ijms24109082_

Round 1
Reviewer 1 Report
The authors used powdery mildew and rust fungi as examples and provided a thorough review on the current status of RNAi technology, especially for functional analysis and controlling diseases caused by these two fungal species. Future technical hurdles that need to be overcome and research directions were also discussed in order for this technology to be used effectively and practically in the disease control. RNAi seems to be an innovative and promising technology, which can be used in both research and disease management. The manuscript was well written in general, but minor revision is required.
L19-20: Suggest delete “for the first time” because gene functions had been studied by other methods, such as gene editing.
L47: “passenger strand” and “guide strand” were mentioned here. It is better to label in Figure 1.
L70-71: Typo “synthesize”, also change it in the figure of “Plant cell” in Fig. 1.
L106: Change “the peculiarities of the biology” to “the biological peculiarities”.
L108: Change to “open up”.
Section “2. Powdery mildew and rust fungi” can be reduced greatly because the disease biology, symptom and fungal life cycle are not the focuses of this review.
Fig. 2B: Should be “appressorium”.
Caption: The Latin name should directly follow the common name, then comes the tissue type, e.g. “Wheat (Triticum aestivum) leaf”.
L173: Wheat has three rusts, Puccinia graminis, P. striiformis and P. triticina.
L303: Change to “Control of pathogen growth occurs”.
L321-327: Suggest break into two sentences. The current sentence is too long.
The following applies to all the tables:
Title: It is better to spell out, rather than using the abbreviations, e.g. spell out “VIGS”.
Plant host: Better be consistent, suggest use Latin names for all, not common names for some and Latin names for the other tables.
Pathogen: If mention the fungal species for the first time in each table, better spell out. “f. sp.” should not be italicized.
Table 1: There should be a border line in between “P. triticina” and “P. graminis”.
L490: Should be “2021”.
L566: Suggest change “will” to “may” because the future of RNAi technology in this area is not for certain and it is just a good possibility and with the further development and improvement, it may become a reality.
The manuscript was well written, only minor editing in English is needed.
Author Response
Dear Reviewer
We appreciate your valuable comments to improve our manuscript. Here are our answers:
L19-20: Suggest delete “for the first time” because gene functions had been studied by other methods, such as gene editing. Done
L47: “passenger strand” and “guide strand” were mentioned here. It is better to label in Figure 1. Labeled.
L70-71: Typo “synthesize”, also change it in the figure of “Plant cell” in Fig. 1. Changed
L106: Change “the peculiarities of the biology” to “the biological peculiarities”. Modified
L108: Change to “open up”. Changed
Section “2. Powdery mildew and rust fungi” can be reduced greatly because the disease biology, symptom and fungal life cycle are not the focuses of this review. This was reduced to the maximum that we could for its correct compression. We would like a public not specialized in these pathogens can know important aspects related with their biology and life cycle.
Fig. 2B: Should be “appressorium”. Changed
Caption: The Latin name should directly follow the common name, then comes the tissue type, e.g. “Wheat (Triticum aestivum) leaf”. Modified
L173: Wheat has three rusts, Puccinia graminis, P. striiformis and P. triticina. This part was deleted to reduce section 2
L303: Change to “Control of pathogen growth occurs”. Changed
L321-327: Suggest break into two sentences. The current sentence is too long. Done
The following applies to all the tables:
Title: It is better to spell out, rather than using the abbreviations, e.g. spell out “VIGS”. Spelled out.
Plant host: Better be consistent, suggest use Latin names for all, not common names for some and Latin names for the other tables. Done
Pathogen: If mention the fungal species for the first time in each table, better spell out. “f. sp.” should not be italicized. Done
Table 1: There should be a border line in between “P. triticina” and “P. graminis”. Included
L490: Should be “2021”. Modified
L566: Suggest change “will” to “may” because the future of RNAi technology in this area is not for certain and it is just a good possibility and with the further development and improvement, it may become a reality. Changed
Many thanks.
Reviewer 2 Report
The review addresses a very relevant topic in the field of plant protection and modern techniques concerning the silencing of gene expression. The authors describe step by step the different possibilities of RNAi use against powdery mildew and rust pathogens. First, they present the general interaction between a plant cell and a fungal pathogen from a molecular point of view and then focus on two important groups of plant pathogens, i.e. powdery mildew and some rust fungi.
The main objective of this study was achieved, because the Authors described in detail how RNAi approach can contribute to the analysis of gene function and how it can present a new strategy for the management of powdery mildew and rust diseases. First, the RNAi were described as a tool for the analysis of genes subjected to silencing via RNA interference in fungal cells, e.g. virus-induced gene silencing (VIGS), host-induced gene silencing (HIGS), Agrobacterium-mediated gene silencing (ATM-HIGS), and direct action of exogenous dsRNA. The possible mitigation of powdery mildew and rust disease by RNAi technology is well-stated and illustrated by many examples. The pros and cons of these techniques have been described.
I have just a few remarks, for a better understanding of reading by non-plant pathogen readers:
- could you develop the term "hpRNA" while the first time used,
- please adjust the style of headlines to the journal rules,
- unify the writing of "ml" or "mL" in Table 6,
- unify the writing style in References, e.g. lines 689-692, 697, 737, 753, 805, 840, and 848.
In conclusion, the manuscript fully presents the possibility of using RNA interference as a promising tool to counteract plants' biotic stressors, i.e. obligate biotrophic pathogens like powder mildew or rust fungi.
Author Response
Dear Reviewer
We would like to thank you for your suggestions, which were very helpful to improve our manuscript. Next, we explained, point-by-point, our responses to your suggestions:
- could you develop the term "hpRNA" while the first time used. Explained
- please adjust the style of headlines to the journal rules, Done
- unify the writing of "ml" or "mL" in Table 6,Done
- unify the writing style in References, e.g. lines 689-692, 697, 737, 753, 805, 840, and 848. We worked on it.
Thank you very much.